# A Bridge to Nowhere: Enabling Autonomy in a Case of Failed ECMO Rescue of Bleomycin-Induced Pulmonary Toxicity

**DOI:** 10.3390/reports6010017

**Published:** 2023-03-20

**Authors:** James Hall, Michael Khilkin, Sara Murphy, George Botros

**Affiliations:** 1Grossman School of Medicine, New York University Langone, New York, NY 10016, USA; michael.khilkin@nyulangone.org (M.K.); george.botros@nyulangone.org (G.B.); 2Columbia University Medical Center, New York, NY 10032, USA; spm9010@nyp.org

**Keywords:** bleomycin, ECMO, pulmonary fibrosis, pulmonary toxicity, veno-venous extracorporeal membrane oxygenation, lung transplant

## Abstract

Extracorporeal membrane oxygenation (ECMO) can be a life-saving intervention in cases of potentially reversible refractory respiratory failure. One such indication can be bleomycin-induced lung injury. However, in some cases, the injury can be so severe that it becomes irreversible and creates complex medical decisions regarding life support and the continuation of care when no additional therapeutic options are feasible, particularly in cases of patients who were young and fully functional prior to an acute illness. In cases of full pulmonary replacement with mechanical support and the degree of functionality that can be attained utilizing modalities such as ECMO can obscure the true severity of illness and make end-of-life decisions significantly harder for families and caregivers.

## 1. Introduction

We present a case of bleomycin-induced pulmonary toxicity in a patient with a testicular germ cell tumor requiring management with extracorporeal membrane oxygenation (ECMO) who was subsequently evaluated for potential lung transplantation. This case emphasizes the importance of informed decision making and the palliative aspects of the case.

## 2. Case Presentation

A 40-year-old male with a past medical history of class I obesity and mild hypertension was diagnosed with a malignant germ cell tumor of the left testis with a 12 cm left retroperitoneal mass and a 5.4 cm lesion in the liver consistent with metastatic disease, staged as N3S2 IIIB. The patient underwent a needle biopsy that showed expression of GATA3, Sal-like protein 4, human chorionic gonadotropin (hCG), and focally epithelial membrane antigen, 34-beta-E12, and Glypican-3 consistent with choriocarcinoma.

He underwent an orchiectomy of the left testis and received a total of four cycles of etoposide, bleomycin, and cisplatin over the ensuing three months. The tumor was responsive to chemotherapy showing a significant decrease in hCG levels from 75,005 IU/L to 149 IU/L and shrinking of the retroperitoneal mass (Figure 1).

One month after finishing the last cycle of chemotherapy, the patient developed shortness of breath and a dry cough and was diagnosed with pneumonia, requiring readmission to the hospital. His initial oxygen saturation on room air was as low as 90%. He was given supplemental oxygen, ceftriaxone, and azithromycin. A CT scan revealed ground glass opacities and air bronchograms and re-demonstrated his retroperitoneal mass and liver lesion, both significantly smaller (Figure 1 and Figure 2).

His oxygenation progressively worsened despite increasing oxygen support over the next week including intubation and mechanical ventilation on hospital day 11. Despite receiving between 55 and 90% oxygen, positive end-expiratory pressure (PEEP) of up to 10 cm’s H_2_O, and neuromuscular blockade, his saturations and pulmonary compliance continued to deteriorate and an ECMO consult was requested on day 6 of mechanical ventilation. The patient was subsequently cannulated for peripheral veno-venous ECMO without incident. His ventilator was adjusted for minimal airway pressures and inspired oxygen, with a PEEP of 10 cm H_2_O, a drive pressure of 15 cm H_2_O, with an FiO_2_ of 30%.

Given the concern for bleomycin toxicity and fibrotic changes, on ECMO day 8, he was treated with pirfenidone. Following a percutaneous tracheostomy on ECMO day 10, he was given a dose of infliximab. Throughout the first several weeks of ECMO support, the patient remained significantly volume overloaded with a weight increase of more than 17 kg despite aggressive diuresis with loop diuretics and a decrease in lean body mass from deconditioning. His sedation and paralytics were gradually weaned until he was significantly more awake. Ventilatory support was reduced and on ECMO day 19, the patient was placed on trach collar which was much more comfortable for him.

However, on ECMO day 19, he suffered an acute desaturation and was found on echocardiography to have acute right heart strain with severe pulmonary hypertension, with a measured pulmonary arterial systolic pressure of 117 mmHg, though the measurement was likely significantly confounded by the flow changes created by the ECMO drainage cannula. He was started on nitric oxide and milrinone for right heart dysfunction resulting in decreased pulmonary pressures and subjectively better right ventricular contractility on echo. His pulmonary function continued to worsen despite antifibrotic therapy as demonstrated by continued bilateral complete opacification of the lungs and progressively decreasing pulmonary compliance which, in the setting of non-escalating inspiratory pressures, resulted in tidal volumes of 50–75 mL and full reliance on the ECMO circuit for oxygenation and carbon dioxide clearance.

According to current guidelines, the patient was not a candidate for transplantation due to his active neoplasm, which is an absolute contraindication for transplant. However, because his beta hCG tumor marker was approaching zero (Figure 3), the treating oncologist believed that his tumor was likely sufficiently treated to achieve an eventual cure. As he had been quite healthy prior to his initial diagnosis, and because the typical outcome for this tumor is near universal cure with treatment and at least one instance of successful pulmonary transplantation has been reported in the literature in a somewhat similar situation, it was deemed reasonable to approach transplant programs about consideration of his candidacy [1]. Potential avenues were aggressively explored by the primary team as well as the pulmonary consultants. In addition, the family independently researched and approached medical centers and transplant teams, often without informing the primary team.

Daily bedside meetings and formal weekly meetings were held with the family and representatives of the primary critical care team and the consulting cardiothoracic, palliative care, pulmonology, and oncology services. The family, of whom multiple members were employed within the health care system, were very engaged and supportive in the meetings and at the bedside. They also seemed to grasp the gravity of the situation and made specific statements reflecting their understanding of the patient’s dire prognosis. However, when the idea of a possible pulmonary transplantation came under consideration, the tone of their comments began to shift, deemphasizing the potential eventuality of a poor outcome and focusing nearly exclusively on an improbably recovery or the chance that a transplant program would deem him to be a suitable recipient. This shift occurred despite clear and often blunt communication from the team that the likelihood of recovery was vanishingly low and that despite our efforts to get the patient evaluated, programs were unlikely to consider him since his putative cancer cure was not provable with the preserve of persistent necrotic masses. Moreover, even if this cancer were not a concern, his general debility and deconditioning would make him a poor candidate for transplant and, indeed, unlikely to survive any significant surgical procedure. Nevertheless, parallel to the team, several family members continued independently to approach numerous transplant programs across the country. The uniform response from these institutions was that the patient would not be an appropriate candidate due to the potential for recurrence of malignancy, including from the program that had previously published a case of transplantation for bleomycin-induced lung injury immediately after chemo and orchiectomy in a patient without metastatic disease or the deconditioning from three months on ECMO [1].

While these discussions were underway, the patient suffered additional complications including, gastroparesis requiring placement of a post pyloric feeding tube, a prolonged episode of shingles, intermittent new atrial arrhythmias, bleeding from tracts made by his ECMO cannulae, and acute renal injury necessitating a brief period of renal replacement from which his kidneys did recover. He responded well to management of these complications, and he was able to be weaned from sedation and hemodynamic support other than ECMO resulting in a gradual improvement in his mental status and alertness. On ECMO day 48, he was able to use a speaking valve with his tracheostomy, and work with physical and occupational therapy. By this time his pulmonary function had become essentially non-existent, and generating tidal volumes of 30–50 milliliters, he was only able to speak one word at a time through the valve.

As his delirium lessened and the family was able to communicate increasingly effectively with him, they insisted that he not be upset with suggestions that he was not recovering. They told him directly that he was improving and that they were finding him a transplant. Initially, it was clear that he had little understanding of the situation or the communication, but as the patient’s cognition improved, the team became increasingly uncomfortable with what seemed to be misleading information. On ECMO day 80, the decision was made to determine whether the patient had the capacity to understand and make decisions for himself. To ensure this understanding, a formal consultation was performed by the psychiatry service, during which a discussion with him was conducted about his current condition, past events, and anticipated prognosis. The attending psychiatrist determined that the patient had full capacity and understood the position he was in. After discussing his condition, treatment, prognosis, and alternatives, the patient insisted that he was very uncomfortable did not want to continue efforts that were unlikely to result in a functional outcome, and would rather be allowed to die. These discussions generated considerable tension with the family as they strongly opposed any transition to comfort measures.

Nevertheless, with established capacity, the patient took a number of days to communicate with various family members and friends and place his affairs in order. At his request, comfort measures, including feedings and discontinuation of uncomfortable procedures, were instituted, and on the day designated by the patient, ECMO day 87, support through the circuit was discontinued, and the patient died.

## 3. Discussion

A number of important clinical and supportive aspects of this case bear emphasis, particularly given the increased use of ECMO over the last several years. These include awareness of and treatment for bleomycin toxicity, the clinical aspects of germ cell tumors and whether transplantation guidelines should be altered, indications for ECMO, and ethical considerations of discontinuing ECMO despite opposition to that withdrawal.

### 3.1. Testicular Cancer

Testicular cancers are typically highly curable neoplasms with a cure rate of >95%. A large proportion of testicular cancers in young men stem from germ cell tumors and can be subclassified into seminomatous or nonseminomatous based on histology. Seminomatous germ cell tumors classically present earlier and have a better prognosis as compared to nonseminomas [2]. An overwhelming majority of seminomas present as stage I disease (roughly 80%), rarely metastasize, and are extremely responsive to radiation therapy. Histopathological diagnosis is typically established with a radical orchiectomy, which is considered first line treatment for localized disease and often results in high cure rates [3]. Further therapy is dictated by the histopathological features, the presence of risk factors, and the extent of disease. For patients who require chemotherapy and have a favorable risk profile, a protocol of three cycles of bleomycin, etoposide, and cisplatin is considered standard therapy [4].

### 3.2. Belomycin Toxicity

Bleomycin is a chemotherapeutic agent derived from *streptomyces verticillus* that is useful in the treatment of multiple neoplasms, including germ cell tumors. Its mechanism of action involves binding to cytosine-guanine rich segments of DNA in the G2 phase of mitosis and causing oxidative damage through the release of free radicals [5]. In addition, bleomycin directly induces lipid peroxidation, causing further oxidative damage. It is this second mechanism that is believed to be involved in the development of the cellular damage, edema, and fibrosis which constitute pulmonary toxicity. This occurs in approximately 3–5% of patients who receive the drug, with higher rates found in patients who have a history of smoking, have received thoracic radiation, or who subsequently receive supranormal levels of inspired oxygen [6,7,8]. Though most cases of pulmonary toxicity occur within the first two weeks after administration, it has been described in the literature as late as five years after its administration [9].

Pirfenidone and glucocorticoids have been used for prevention and treatment of bleomycin-associated fibrosis with some reported success, though randomized trials have not yet been reported [10,11]. With treatment, mortality remains at 10–20% [10]. Veno-venous ECMO has been used for rescue in the treatment of bleomycin toxicity, with published rates of successful decannulation of 66% [12].

### 3.3. Lung Transplant in the Setting of Recent Neoplasm

As has long been documented in the medical literature, immunosuppressive regimens are associated with de novo neoplasm or recurrence of prior disease, and all solid lung transplants require robust immunosuppressive regimens [13].

Though immunosuppressive regimens for pulmonary transplantation are among the strongest, thereby implying the highest risk for neoplasm, a recent analysis of pulmonary transplant recipients from the International Society of Heart and Transplant registry of more than 13,000 patients showed no increased risk of five-year mortality for those with malignancy prior to transplantation [14]. Existing guidelines for all solid organ transplants were extrapolated from limited data in renal transplant registries. In a recent consensus statement, the American Society of Transplantation advocated for reassessing these long-standing guidelines for solid organ transplants in the setting of a prior malignancy, traditionally requiring a two to five year cancer free interval prior to consideration [15]. Though they did not specifically examine choriocarcinomas, based on a reported low recurrence risk of 0–10% with testicular cancers, other neoplasms with similar rates of recurrence were assigned the recommendation of waiting one to two years post therapy prior to transplantation [15,16]. While expert opinion would be required to weigh in on specific guidelines, similar reasoning could be used to justify early transplantation, potentially without any waiting period for patients in whom confidence of cure could be assured by therapeutic and surgical interventions, as was the case in the patient reported on by Narayan et al. who had suffered irreversible pulmonary injury from bleomycin toxicity requiring ECMO following treatment for testicular cancer and underwent successful lung transplantation only five weeks after completing chemotherapy. In this case, the tumor was not metastatic, and treatment along with orchiectomy was considered curative [1]. Similarly, liver transplants are currently being performed for patients with colon cancer with unresectable liver metastases when there is a high degree of confidence in surgical cure [13].

Unfortunately, regardless of oncologic concerns, numerous other factors influence outcomes after organ transplantation including age, comorbidities, and functional status, and solid organs remain a scarce resource. With respect to our patient, while we were not able to surgically assure the absence of cancer, his complications, particularly his right heart dysfunction and generally weak condition independently made him a poor candidate for transplant.

### 3.4. ECMO Withdrawal and Ethical Considerations

It is the nature of intensive interventions to cause pain and potentially lead to complications by creating artificial aberrations of normal physiologic processes with the goal of prioritizing vital functions over recoverable injuries to create a reasonable path to what may be considered a favorable outcome. ECMO originated as an adaptation of cardiopulmonary bypass in the operating room suitable for longer term use outside the operating theater. Although it is different in many ways, it also is intended to provide transient support only—simply put, the goal of ECMO is to serve as a bridge to recovery or transplantation. When there is no chance of recovery or transplantation, such as in the case of our patient, ECMO becomes a bridge to no recovery. As a very uncomfortable, invasive, and dangerous intervention, ECMO is justifiable only when prognoses are dire, alternative options either do not exist or are potentially more harmful, and when there is a reasonable path to a desirable outcome. ECMO is considered when predicted mortality exceeds 50% and indicated when it exceeds 80%, a prediction often obtained by applying the Murray Score for acute lung injury [17,18]. Given the high rates of complications with ECMO and the large proportion of people who do not survive, discussions with the patient, if possible, or other decision makers about reasonable expectations of success and endpoints if improvements are not achieved are vital prior to committing the patient to the risks and discomforts of the intervention. In addition, the possibility that ECMO becomes a bridge to no recovery must be addressed. This is a particularly challenging situation for patients, families, and care teams because while on ECMO, the patient is fully supported, with the function of certain vital systems replaced, allowing improvement and even normalization of other organ systems, including cognition [19]. Withdrawing medications, ventilators, and dialysis because of clinical improvements can create a false sense of hope that the primary problems requiring ECMO may also have been overcome, even in the face of clear and consistent communication to the contrary. When, as is often the case, this situation arises in patients who have not regained cognition and remain obtunded or unconscious, the burden of end-of-life decisions is placed on the families, whose decisions are made substantially more difficult in the setting of apparent clinical improvement.

The principles of bioethics as set forth by Beauchamp and Childress in 1985, now widely accepted as the standard of care, are autonomy, beneficence, nonmaleficence, and justice [20]. These principles are non-hierarchical and should each be weighed when making ethical clinical decisions. Respecting the principle of autonomy requires that discussions be frank with as high a degree of familial understanding of clinical considerations and medical team understanding of individual values as can be achieved in order to avoid situations in which continuation of ECMO support results only in further harm without any expectation of benefit. When such discussions fail, tension can develop between families and care teams.

Failure to reach consensus, often interpreted as poor communication between the family and healthcare team, is one of the leading causes of Post-Intensive Care Syndrome-Family (PICS-F), a term coined in 2010 to describe the adverse mental health outcomes experienced by close family members of ICU patients [21]. PICS-F encompasses a variety of symptoms, the most common being sleep deprivation, anxiety, depression, and complicated grief which are often associated with decreased employment and social or relationship difficulties for years following the ICU experience [22]. The importance of family involvement in improving the care of critical patients is well established and widely accepted. Less acknowledged, is the importance of the care provided directly to the family to support them during difficult situations. It is important to consider the impact of severe illness and intensive interventions to the family in order to mitigate the effects of PICS-F. A multi-disciplinary approach to care with regular family meetings has been shown to decrease the effects of PICS-F and should be incorporated into care [23,24].

### 3.5. Resolution

In the case of our patient, regular multidisciplinary meetings were held and the impossibility of a surviving outcome were clearly communication by all members of the team. Nevertheless, unrealistic expectations persisted among family members with respect to patient prognosis and the availability of transplant as an intervention. The absence of an irreversible acute decompensation also facilitated the impression of an indefinite time frame in which some other solution could be discovered. This dissonance was resolved only when the patient himself regained decision making capacity, was able to understand the situation clearly, and then communicate clearly his wish to discontinue therapy despite the feelings of his family. Following this expression, the family’s disagreement primarily rested on the question of the patient’s capacity. They believed that the patient’s recent delirium prevented his understanding and therefore impair his ability to make end-of-life decisions. In an effort to address their concerns while respecting autonomy, psychiatry was involved to formally assess the patient to make the determination of capacity. This attempt, however, failed to resolve his family’s concerns. Ultimately, they did not object to his capacity itself, though continued to express skepticism, but rather strongly opposed his choice to discontinue ECMO. Ultimately, no consensus was achieved between the patient and the family. Failure to reach consensus, often interpreted as poor communication between the family and healthcare team, remains one of the leading causes of PICS-F. This raises the question of how much latitude to give the family in order to reach consensus while still respecting the patient’s autonomy.

## 4. Conclusions

Complicated treatment regimens which include significant artificial support may mask the severity of illness and impair the ability of designated decision makers to accurately assess the situation. This may lead to significant disagreement with health care providers. Clear expectations should be communicated and contingency measures in the event of a no-recovery-situation should be established prior to the initiation of this support, when possible. Whenever feasible, the best decision maker for a patient is the patient themselves, but in the setting of critical illness, particularly when irreversible decisions are required, the capacity of the patient to understand the situation and communicate their wishes must be clearly established. Disagreements between families, the patient, and/or the healthcare team may lead to tension that can further impair communication and potentially compromise care and well as contribute to lasting harm to survivors’ mental health.

## Figures and Tables

**Figure 1 reports-06-00017-f001:**
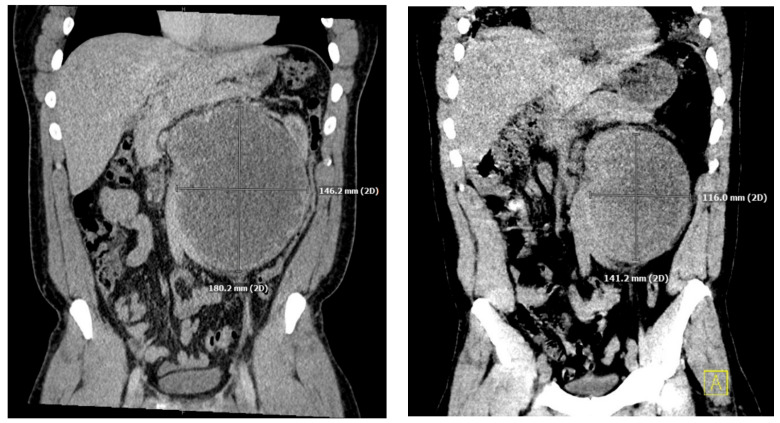
Coronal slices of CT images performed immediately after diagnosis of retroperitoneal mass (**left**) originally measuring 18 × 14 cm, and three months after initiation of treatment (**right**) measuring 14 × 12 cm.

**Figure 2 reports-06-00017-f002:**
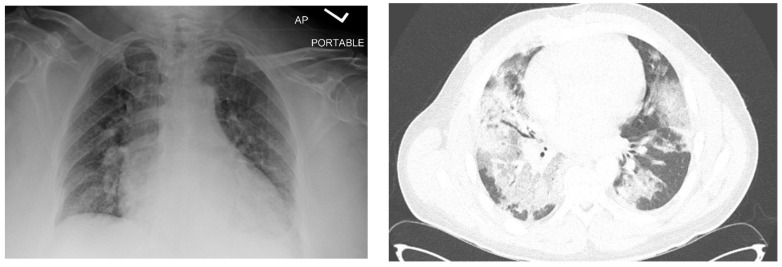
Chest radiograph (**left**) and axial image from chest CT (**right**) showing patchy opacifications on the day of admission.

**Figure 3 reports-06-00017-f003:**
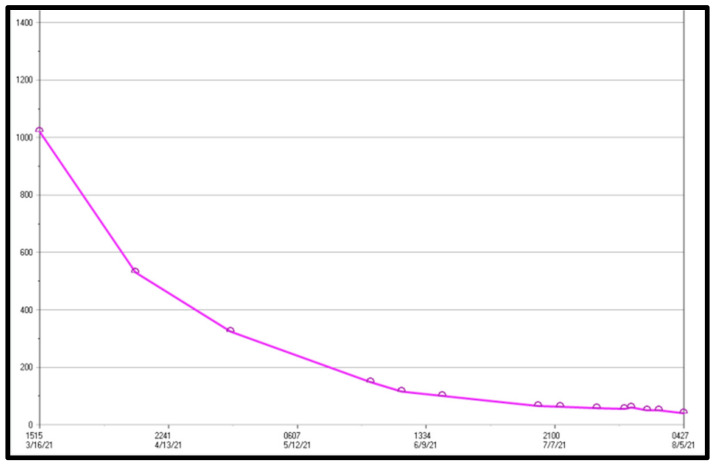
Beta hCG trend from one month post initiation of chemo to day 60 of ECMO support.

## Data Availability

Data sharing is not applicable to this article.

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
