# Peer review of "A Bridge to Nowhere: Enabling Autonomy in a Case of Failed ECMO Rescue of Bleomycin-Induced Pulmonary Toxicity"

_reports, 2023, doi:10.3390/reports6010017_

Round 1

Reviewer 1 Report

As you mentioned in the abstract, end-of-life decisions of patients using ECMO are significantly harder for families and caregivers.

It can be evaluated that this article describes the acceptance of the family until the end and analyzes the causes of conflicts .

However, the probability of clear communication prior to initiation of ECMO written in the conclusion may not be described sufficiently in 3.5 Resolution.  This point should be corrected. 

Reviewer 2 Report

reports-2280651

The authors present a case report of an extremely specific medical case in a patient with a testicular germ cell tumor.  The manuscript is well presented and just some stylish minor spell check is required.

A few suggestions:

1. In the topic to be: "Bleomycin-induced"

2. Font unification; Fig.1 and Fig.2 and in the reference part -there are uncharacteristic highlights;

3. The conclusion needs to be improved.
